# Sexual assault cases at a tertiary referral hospital in urban Ethiopia: One-year retrospective review

**Lemi Belay Tolu*, Wondimu Gudu**

Department of Obstetrics and Gynecology, Saint Paul's Hospital Millennium Medical College, Addis Ababa, Ethiopia

* lemi.belay@gmail.com

**Data Availability Statement:** All relevant data are within the paper and its Supporting Information files.

**Funding:** The author(s) received no specific funding for this work.

## Abstract

### Introduction

Sexual assault is an important health and social problem affecting young girls. The aim of the study is therefore to do a one-year retrospective review of documents of alleged sexual assaults managed at Saint Paulo's Hospital Millennium Medical College (SPHMMC) to determine survivors' characteristics, circumstances of the assault, and treatment offered.

### Methods

This is a hospital-based one-year retrospective review of alleged sexual assault cases. The case records of survivors were retrieved, reviewed and information extracted analyzed using SPSS version 17. Characteristics of victims of the sexual assault, clinical presentation, and management provided were described by frequency and percentage distribution.

### Result

A total of 170 cases of alleged sexual assault who received care during the study period were identified. Around 96% of the survivors were female while there were 6 male cases. The mean age of the victims was 13 yrs. with a range from 2 to 25 yrs. About 23.6% of the victims were less than 10 years. Half of the victims were assaulted by neighbors (45%) followed by strangers (36.5%). The interval between the incident and presentation to the hospital ranged from 2 hours to 93 days (2224 hours) with an average of 98 hours. Most (93.0%) had one or more physical examination findings at presentation. Serology tests for HIV, Hepatitis B, and Syphilis were done in 97.3%, 88.7%, and 84.5% cases respectively. Urine pregnancy tests were done in 62.5% of the cases. Prophylaxis against HIV and STI prophylaxis was provided to 42% and 45% respectively. Social support/counseling was provided to 61% of the victims and legal evidence (certificate) was provided to 45.5% of the cases.

### Conclusion and recommendations

Although it is largely not reported by the victims, sexual assault is a grievous offense still happening constantly. Children and young girls remain the most vulnerable. There is

**Competing interests:** The authors have declared that no competing interests exist.

**Abbreviations:** SPHMMC, Saint Paulo's Hospital Millennium Medical College; SPSS, Statistical package of social science; STI, Sexual Transmitted Infections; OPD, Out Patient Department; PEP, Post Exposure Prophylaxis; EC, Emergency Contraception; RTS, Rape Trauma Syndrome; WHO, World Health Organization.

inadequate forensic evidence collection, legal and medical care. There is also a delay in presentation to hospital by victims. Therefore, there is a need to have standardized protocols for comprehensive evaluation and care of the survivors. It is also imperative that a multidisciplinary approach like a one-stop clinic should be utilized to provide effective and efficient medical, social, psychological, and legal services. Finally, it is very necessary to increase public awareness and preventive interventions are required particularly to protect the vulnerable age group to enhance their safety.

## Introduction

The World Health Organization(WHO) defined sexual assault as a spectrum of activities ranging from rape to physically less intrusive sexual contacts, whether attempted or completed [1–3]. Accordingly, rape is not a medical diagnosis. It is a legal terminology reserved for cases of penile penetration of the victim's vagina, mouth, or anus without consent [2, 3]. Other types of sexual assault and rape spectrums include forced or coerced vaginal or anal penetration by any body parts or object; breast or genitalia fondling or being forced or coerced to touch another person's genitalia [3]. It involves a lack of consent; the use of physical force, coercion, deception, or threat; and/or the involvement of a victim that is mentally incapacitated or physically impaired (due to voluntary or involuntary alcohol or drug consumption), asleep or unconscious [3].

Sexual assault is not peculiar to any race or socio-economic class, although it is largely hidden by the victims it is estimated that 12 million people around the world face sexual violence every year [4]. Young people are the most frequent victims of sexual violence; it is generally thought that 12% to 25% of girls and 8% to 10% of boys under 18 years of age will suffer sexual violence [5, 6].

The World Health Organization reports that one in every five women is a victim of sexual assault [7] and globally, 35% of women have experienced either physical and/or sexual intimate partner violence or non-partner sexual violence [4]. The regions of the world with the highest reported rates of sexual and physical violence towards women are Africa, the Middle East, and Southeast Asia [4]. In Africa, 5–15% of the females report a forced or coerced sexual experience [8]. In South Africa, the prevalence of rape, from community-based reports shows a figure of 2070 per 100,000 per year. Reports from Ethiopia showed that from 367 high school girls about 33.3% of the participant's first intercourse was rape [9]. A cross-sectional study of sexual assault cases at two hospitals in Addis Ababa revealed that more than half of the victims were children and adolescents [10]. Another study from Ethiopia conducted in Addis Ababa reported that among crimes committed against children 23% of them were child sexual victimization [11] Evidence from the USA also shows that children and adolescents have the highest rates of rape and other sexual assaults of any age group [12].

Sexual assault is a traumatic experience that occurs across all societies and disproportionally affects adolescent and young adult women [13] and is often associated with psychological, physical, and social distress [14]. Assailants are known to their victims who perpetrate this act during the daytime and survivors often delay in seeking care [5, 15, 16]. Exposure to sexual violence is associated with a range of health consequences for the victim so timely report and comprehensive care must address physical injuries; pregnancy; STIs, HIV and hepatitis B; counseling and social support; and follow-up consultations because the longer the delay in a hospital report the lower the quantity and quality of forensic evidence [3, 17], and the higher the risk of negative health outcomes.

Sexual assault may vary with time and from place to place in the country. Though it is one of the sexual assault treatment centers in Addis Ababa, the capital city of Ethiopia we couldn't find a recent documented study from Saint Paul's Hospital Millennium Medical College (SPHMMC). The aim of the study is therefore to do a one-year retrospective review of documents of alleged sexual assaults managed at SPHMMC to determine survivors' characteristics, circumstances of the assault, and treatment offered to suggest possible prevention strategy to reduce the incidence as well as improving evaluation and management approaches of survivors.

## Methods and materials

This is a hospital-based one-year retrospective review undertaken from October 1, 2018, up to October 1, 2019, at Saint Paul's Hospital Millennium Medical College in Addis Ababa, Ethiopia. Saint Paul's hospital is one of the tertiary referral hospitals and a teaching hospital for the Millennium Medical College. Service for sexual assault victims is provided at the Michu clinic which is found at the frontier of the hospital at the Emergency gate and is a dedicated clinic for abortion, family planning, and sexual assault service.

The source population was all women getting reproductive service, whereas the study population was all alleged sexual assaults managed at Michu clinic of Saint Paul's Hospital Millennium Medical College. All alleged sexual assault cases registered at Michu clinic were included except those with incomplete documentation of important variables.

The medical records of sexual assault cases managed at Michu clinic were approached for the identification of alleged sexual assault cases who received medical care within the study period. A sexual assault case was defined as any person, irrespective of age reporting any type of non-consensual sexual activity whether attempted or completed. The case records of survivors were then retrieved from the record office by data collector nurses, reviewed and information extracted was entered into pretested questionnaires that evaluate the socio-demographic characteristics, place and time of the incident, the relationship of assailants to victims, methods employed by the assailant to overcome victims, forensic specimen collection, the treatment offered and follow up of survivors.

Data collectors were trained in pre-tested checklists and principal investigators cross-checked for completeness and accuracy of the collected data regularly.

Data were exported and analyzed using SPSS version 17. Frequency and percentage distribution of characteristics of victims of the sexual assault were calculated in terms of age, sex, place of residency, marital status, occupation, level of education, place and time of the incident, the relationship of assailants to victims, methods employed by the assailant to overcome victims and forensic specimen collection. The sexual assault victim clinical presentation and management provided were also described by frequency and percentage distribution.

## Operational definitions

### Sexual violence

Any sexual act, attempt to obtain a sexual act, unwanted sexual comments or advances, or acts to traffic a person's sexuality, using coercion, threats of harm or physical force, by any person regardless of relationship to the survivor, in any setting, including but not limited to home and work environments.

### Rape

An act of non-consensual sexual intercourse including the invasion of any part of the body with a sexual organ and/or the invasion of the genital or anal opening with any object or body part.

### Attempted rape

Efforts to rape someone which do not result in penetration.

### Sexual abuse

Other non-consensual sexual acts, not including rape or attempted rape and includes acts performed on a minor.

### Sexual assault

A major form of sexual violence that includes at least rape, attempted rape, and sexual abuse.

### Perpetrator/assailant

A male or female, group or institution that inflicts, supports, or condones violence or other abuses against a person or group of persons.

### Survivor/victim

A person who has lived through an incident of sexual assault. Survivor is a more preferred term as it has a positive connotation.

### Ethical consideration

Ethical clearance & permission letter to conduct the study and publish the outcome was obtained from the Institutional Review Board (IRB) of SPHMMC. In this retrospective study, the ethics committee waived written consent and we decided not to contact victims for permission months or the year after the incident. We included a patient who sought care for sexual assault between October 1, 2018, up to October 1, 2019. We collected data for two months from December 1, 2019, to January 30, 2020. Confidentiality was maintained during data collection, analysis, and interpretation and all data were fully anonymized before accessing them. Furthermore, we did not record any patient identifier and returned client records to its place after the completion of data collection. All the datasets used and/or analyzed during the current study were included in the manuscript.

## Result

In this study, there were a total of 170 cases of alleged sexual assault who received care at SPHMMC were included after excluding 5 records for missing important variables. Around 174 (96%) of the survivors were female while there were 6 (4%) male cases. Most were single 169 (99.4%) and students 130 (76.5%). The mean age of the victims was 13 years with a range from 2 to 25 yrs. Forty (23.6%) of the victims were less than 10 years (Table 1).

Regarding the circumstance of the assault, almost half of the victims were assaulted by neighbors (45%) followed by strangers 62(36.5%). Multiple assailants were involved in 11 (6.4%) of the cases. Most (60%) of the incidents did occur during the daytime and physical force and weapons were used in 74 (43.5%) and 41 (24%) of the incidents respectively. The interval between the incident and presentation to the hospital ranged from 3 hours to 93 days with an average of 98 hrs. (Table 2).

Among the 113 (66%) female victims who had body orifice penetration, genital penetration was reported in 109 (64%); rectal in 9 (5%), and oral in one of the cases whereas 5 (3%) had multiple orifice penetration. Among 6 male victims, five of them had anal penetration

**Table 1. Sociodemographic characteristics of the victim of sexual assault cases at SPHMMC from October 1, 2018, up to October 30, 2019.**

| Characteristics (n = 170) | No. | % |
|---|---|---|
| **Age** | | |
| ≤10 years | 40 | 23.6 |
| 10–19 years | 100 | 58.8 |
| ≥ 20 years | 30 | 17.6 |
| **Sex** | | |
| Male | 6 | 3.5 |
| Female | 164 | 96.5 |
| **Level of education** | | |
| None | 30 | 17.6 |
| Primary school | 76 | 44.7 |
| High school | 55 | 32.4 |
| College /university | 9 | 5.3 |
| **Occupation** | | |
| Preschool | 34 | 20 |
| Student | 130 | 76.5 |
| Unemployed | 1 | 0.6 |
| Housewife | 3 | 1.8 |
| Domestic servant | 1 | 0.6 |
| Others | 1 | 0.6 |
| **Marital status** | | |
| Single | 169 | 99.4 |
| Married | 1 | 0.6 |
| **Address** | | |
| Addis Ababa | 127 | 75.0% |
| Out of Addis | 43 | 25.0% |

(attempted/complete). Ejaculation in the victim's genitalia or elsewhere on body part was reported in 80 (47.0%) of the cases and 78(98%) of the assailants didn't use condoms.

The finding of clues/evidence for a possible sexual assault is affected by what the victims do after the assault. In this study, bathing and douching were the commonly reported practices in 103 (60%) and 68 (52%) respectively (Table 3).

The clinical presentation of the sexual assault survivors was variable. Most (93.0%) had one or more physical examination findings at presentation, the commonest being genital injury in 112 (72%), vaginal bleeding in 50 (30.5%), and genital discharge in 49 (30%) (Table 4). Of those survivors in whom the genital injury was documented, around 71 (43.3%) had hymenal injuries. Of these 19 (26.7%) had tears at multiple sites with the commonest site of injury is at the 12 o'clock position. The hymenal injuries were described as tears in 62 (87.0%) and abrasions in 9 (13%). An apparent erythematous vestibule/labium was recorded as a possible sign of assault in 6 cases. There were 5 cases of anal injuries among female survivors which were all minor lacerations without anal sphincter injury. The physical findings in all five male victims who reported anal penetration were perianal lacerations with intact anal sphincter. Sixteen of the cases have associated non-genital body injuries most of them sustaining facial laceration.

There are recommended minimum laboratory tests to be done according to the national protocol on the management of victims of sexual assault. In this study, all the recommended minimum laboratory investigations were done in only 120 (71.0%) of the cases.

**Table 2. Circumstance and reporting of the assault cases at SPHMMC from October 1, 2018, up to October 30, 2019 (N = 170).**

| Circumstances of the assault (n = 170) | No. | % |
|---|---|---|
| **Reporting time to hospital** | | |
| ≤ 72 hrs. | 48 | 28.3 |
| 72 hrs-6 days | 50 | 29.4 |
| 7 days-1 months | 42 | 24.7 |
| ≥ 1 month | 30 | 17.6 |
| **Brought to hospital by** | | |
| Family | 147 | 86.5 |
| Police after reporting | 4 | 2.4 |
| Assailant | 1 | 0.6 |
| Victim by herself/himself | 18 | 10.6 |
| **Relation with assailants** | | |
| Neighbors | 76 | 44.7 |
| Stranger | 62 | 36.5 |
| Acquaintances | 11 | 6.5 |
| Family members | 9 | 5.3 |
| Boyfriend | 12 | 7.1 |
| **Number of assailants** | | |
| Single | 159 | 93.5 |
| Multiple | 11 | 6.5 |
| **Number of a sexual assault incident** | | |
| Once | 145 | 85.3 |
| Multiple times | 25 | 14.7 |
| **Place of the assault (s)** | | |
| Neighbors house | 60 | 35.3 |
| Street corner | 14 | 8.2 |
| Friends' homes | 8 | 4.7 |
| Victims home | 37 | 21.8 |
| Uncompleted building | 1 | 0.6 |
| Assailants home | 29 | 17.1 |
| Forest | 5 | 2.9 |
| Others | 16 | 9.4 |
| **Time of the incident** | | |
| Day | 102 | 60 |
| Night | 68 | 40 |
| **Victim mental illness** | | |
| Yes | 6 | 3.5 |
| No | 164 | 96.5 |
| **Did assailants use physical force?** | | |
| Yes | 74 | 43.5 |
| No | 96 | 56.5 |
| **Use of weapons and restraints** | | |
| Yes | 41 | 24.1 |
| No | 129 | 75.9 |
| **Use of medications/drugs/alcohol/inhaled substances by the assailant** | | |
| Yes | 9 | 5.3 |
| No | 161 | 94.7 |

Serology test for HIV, Hepatitis B and Syphilis were done in 146 (97.3%),134 (88.7%) and 127 (84.5%) respectively. A urine pregnancy test was done in 102 (62.5%) of the cases. No Blood or mouth wash samples for DNA were taken. Only in a single case a mouth cotton swab or washing was done. Only 58 (34%) of the survivors had a genital/anal swab taken for the demonstration of sperm cells.

The laboratory test results are non- revealing in all the cases except in 2 cases who had a positive test for HBsAg and 5 who had a positive pregnancy test. Among those who had Vaginal and anal swabs analyzed evidence of sperm cells was reported in 23(39.2%) of the cases and signs of genital infection in three. Emergency contraception, Post Exposure Prophylaxis against HIV, and STI were provided to 63 (42%), 68 (45%), and 92 (61.7%) respectively. Social support/ counseling was provided to 78 (61%) of the victims. Legal evidence (certificate) was provided to 58 (45.5%) of the cases. Among those survivors who were provided with emergency contraception, post-pill (Levonorgestrel) was the commonest method provided in 40 (63%) followed by Combined high dose Oral contraceptives in 16 (25%) of the cases. The commonest STI prophylaxis regimen prescribed to the victims was single doses of the combination of ceftriaxone, azithromycin, and metronidazole. Termination of pregnancy was provided for five of the pregnant survivors. Follow up visits were made in 64 (50%) of the survivors (Table 5).

## Discussion

In the current study, there were a total of 170 cases of alleged sexual assault who received care at SPHMMC from October 1, 2018, up to October 1, 2019. The mean age of the victims was 13 years with a range from 2 to 25 yrs. About 23.5% of the victims were less than 10 years of age which is less than other similar study reports [10, 11, 18, 19]. The average duration of the presentation to the hospital was 98 hours. The delayed presentation is because of the high percentage of children and teenagers in the current study who might not report until the parents notice the condition. For relatively older victims fear of stigmatization might result in a delay in reporting. This is a prolonged time compared to the recommended timing of the provision of medical care within 72 hours after the incident. However, the current study finding is similar to other study reports from Africa [18, 19].

The current study reveals that almost half of the victims were assaulted by neighbors followed by strangers in 36.5%, which explains why 35.3% of the assault occurs in the neighbor's house. In this condition the possibility of being caught is slim. It also explains the reason why most (60%) of the incidents did occur during the daytime and physical force and weapons were used to overcome resistance from victims. Besides, most of the victims in the current review were children and teenagers, and the fact that most incidents happened in neighbor's houses might explain more likelihood of daytime occurrence when family members were not around. The finding is similar to the 5-year retrospective study of cases of sexual assault in Lagos, Nigeria in which neighbors were assailants in 54.9% of cases [18].

Most (93.0%) had one or more physical examination findings at presentation, the commonest being genital injury in 72%, vaginal bleeding in 30.5%, and genital discharge in 30%. Of those survivors in whom the genital injury was documented, around 43.3% had hymenal injuries. This is explained by the fact that 60% of the assailants used physical forces in the current review which might result in physical injury. The finding is similar to a medicolegal study of domestic violence in the southern region of Jordan by Hasan Al-Hawari and Asmaa El-Banna [20]. The physical findings in all five male victims who reported anal penetration were perianal lacerations with intact anal sphincter.

In this study, all the minimum recommended laboratory investigations were done in only 71.0% of the cases. Serology tests for HIV, Hepatitis B, and Syphilis were done in 97.3%,

**Table 3. Body orifice penetration and measures are taken by a survivor of sexual assault before reporting to the hospital.**

| Measures taken by alleged sexually assaulted | No. | % |
|---|---|---|
| **Body orifice penetration** | | |
| Yes | 113 | 66 |
| • Genital | 109 | 64 |
| • Rectal | 9 | 5 |
| • Oral | 5 | 3 |
| No | 57 | 34 |
| **Condom use (n = 80)** | | |
| Yes | 2 | 2 |
| No | 78 | 98 |
| **Bathing (n = 170)** | | |
| Yes | 103 | 60.6 |
| No | 67 | 39.4 |
| **Douching (n = 164)** | | |
| Yes | 86 | 52.4 |
| No | 78 | 47.6 |
| **Wiping (n = 170)** | | |
| Yes | 79 | 46.5 |
| No | 91 | 53.5 |
| **The use of tampons (n = 164)** | | |
| Yes | 80 | 48.8 |
| No | 84 | 51.2 |
| **Changes in clothing (n = 170)** | | |
| Yes | 105 | 61.8 |
| No | 65 | 38.2 |
| **Brushing of a tooth (n = 170)** | | |
| Yes | 84 | 49.4 |
| No | 86 | 50.6 |

88.7%, and 84.5% cases respectively. Urine pregnancy tests were done in 62.5% of the cases. Care providers should offer these medical investigations to all eligible victims. However, it is better than previous study findings from Ethiopia and other Africa countries [18, 19, 21]. Two cases had a positive test for HBsAg, and 5 cases had a positive pregnancy test in the current review. About 34% of the survivors had a genital/anal swab taken for the demonstration of sperm cells. No other forensic samples were taken among the cases included in this review. Among those who had high vaginal and anal swabs analyzed evidence of sperm cells were reported in 39.2% of the cases and signs of genital infection in three of the cases. This might be explained by e delayed presentation of the cases to hospitals. The finding of clues/evidence for a possible sexual assault is also affected by what the victims do after the assault. In the current study, almost half of the victims changed cloth, used tampons, brushed tooth, bathed, and did douche before presentation to the hospital. Lack of adequate forensic samples and findings from simple swabs not only hinder justice but also promote non-disclosure and encourage the perpetuation of rape.

Among victims who presented within 72 hours of emergency contraception, Post Exposure Prophylaxis against HIV and STI prophylaxis was provided to 42%,45%, and 61.7% respectively. This is slightly better than the previous study from Ethiopia in which emergency contraception

**Table 4. Clinical presentation among victims of sexual assault at SPHMMC from October 1, 2018, up to October 30, 2019.**

| Clinical presentations | No. | % |
|---|---|---|
| **Genital discharge (n = 164)** | | |
| Yes | 49 | 29.9 |
| No | 115 | 70.1 |
| **Urinary symptoms(N = 170)** | | |
| Yes | 16 | 9.4 |
| No | 154 | 90.6 |
| **Anal pain(N = 170)** | | |
| Yes | 8 | 4.7 |
| No | 162 | 95.3 |
| **Anal bleeding(N = 170)** | | |
| Yes | 4 | 2.4 |
| No | 166 | 97.6 |
| **Abdominal pain(N = 170)** | | |
| Yes | 21 | 12.4 |
| No | 149 | 87.6 |
| **Genital bleeding or discharge (n = 164)** | | |
| Yes | 50 | 30.5 |
| No | 114 | 69.5 |
| **No genital physical injury (N = 158)** | | |
| Yes | 38 | 24.1 |
| No | 120 | 75.9 |
| **Genital injury(N = 156)** | | |
| Yes | 112 | 71.8 |
| No | 44 | 28.2 |
| **Anal injury(N = 151)** | | |
| Yes | 5 | 3.3 |
| No | 146 | 96.7 |

provision was only 25% [22]. In the current review, social support/counseling was provided to 61% of the victims and legal evidence (certificate) was provided to 45.5% of the cases. This indicates that more than half of them did not get appropriate legal services which encourage the perpetuation of sexual assault. No formal consultation or referral for psychotherapy was done in this review. Follow up visits were made in only 50% of the survivors. This is similar to other study findings from Ethiopia and other Africa countries [18, 19]. Psychological impact including rape trauma syndrome can happen in the acute period and the long run after the incident [23, 24]. Therefore, appropriate psychotherapy and follow up arrangements should be offered to all victims to moderate and mitigate the negative health effects on victims.

This review has its limitations. The hospital-based nature of this study might limit the generalizability of its findings to the larger population. Besides, being a retrospective review, the review was also constrained by the availability of data in the case records.

## Conclusions and recommendations

### Clinical implications

Although it is largely not reported by the victims, sexual assault is a grievous offense still happening constantly. Children and young girls remain the most vulnerable. Most of the victims

**Table 5. Type of investigations performed for victims of alleged sexual assault (N = 151).**

| Investigations performed | No* | % |
| --- | --- | --- |
| **Genital cotton swab and slide** | | |
| Yes | 60 | 39.7 |
| No | 91 | 60.3 |
| **Mouth cotton swab or washing** | | |
| Yes | 1 | 0.7 |
| No | 150 | 99.3 |
| **Anal cotton swab** | | |
| Yes | 1 | 0.7 |
| No | 150 | 99.3 |
| **Blood or urine for toxicology** | | |
| Yes | 10 | 6.6 |
| No | 141 | 93.4 |
| **Blood/mouthwashes for DNA** | | |
| No | 151 | 100 |
| **Clothing or skin cotton swab for foreign body** | | |
| Yes | 6 | 4 |
| No | 145 | 96 |
| **Pregnancy test** | | |
| Yes | 102 | 67.5 |
| No | 49 | 32.5 |
| **VDRL** | | |
| Yes | 127 | 84.1 |
| No | 24 | 15.9 |
| **HBsAg** | | |
| Yes | 134 | 88.7 |
| No | 17 | 11.3 |
| **STI screenings** | | |
| Yes | 76 | 50.3 |
| No | 75 | 49.7 |
| **HIV test** | | |
| Yes | 146 | 97.3 |
| No | 4 | 2.7 |
| HIV prophylaxis | 68 | 45 |
| STI prophylaxis | 92 | 61.7 |
| Social support/counseling | 78 | 61 |
| Legal certificate | 58 | 45.5 |
| Emergency contraception | 63 | 42 |

*-only the number of records with complete data is included and the denominator is adjusted accordingly.

were assaulted by neighbors at the neighbor's house during day time. There is inadequate forensic evidence collection, legal and medical care. There is a significant delay in presentation to hospital by victims. Additionally, most victims changed cloth, used tampons, brushed tooth, bathed, and did douche before presentation to the hospital. These factors affecting evidence for a possible sexual assault not only hinders justice but also promotes non-disclosure and encourage the perpetuation of rape. Therefore, there is a need to have standardized protocols for comprehensive evaluation and care of the survivors. It is also imperative that a multidisciplinary approach like a one-stop clinic should be utilized to provide effective and efficient medical,

social, psychological, and legal services. Finally, it is very necessary to increase public awareness and preventive interventions are required particularly to protect the vulnerable age group to enhance their safety.

## Research and policy implications

The current study is limited to one hospital and retrospective in nature. However, the findings shed light on awareness of the society on seeking care for sexual assault and the quality of care provided to survivors. There is a need to conduct a national survey on the quality of care provided to sexual assault survivors. Furthermore, it is very necessary to evaluate the performance of existing policies and programs towards sexual assault to develop contextual policies and guidelines. Finally, we would like to recommend that there should be monitoring and evaluation of preventive and management approaches to sexual assault.

## Supporting information

**S1 Checklist. Describes a completed strobe checklist for an observational study.** (DOCX)

## Acknowledgments

We thank Saint Paulo's Hospital Millennium Medical College Gynecology and Obstetrics and Forensic pathology department staff in general and Michu Clinic staff for helping us with the data collection process.

## Author Contributions

**Conceptualization:** Lemi Belay Tolu, Wondimu Gudu.

**Data curation:** Lemi Belay Tolu, Wondimu Gudu.

**Formal analysis:** Lemi Belay Tolu, Wondimu Gudu.

**Investigation:** Lemi Belay Tolu, Wondimu Gudu.

**Methodology:** Lemi Belay Tolu, Wondimu Gudu.

**Project administration:** Lemi Belay Tolu, Wondimu Gudu.

**Resources:** Lemi Belay Tolu, Wondimu Gudu.

**Software:** Lemi Belay Tolu, Wondimu Gudu.

**Supervision:** Lemi Belay Tolu, Wondimu Gudu.

**Validation:** Lemi Belay Tolu, Wondimu Gudu.

**Visualization:** Lemi Belay Tolu, Wondimu Gudu.

**Writing – original draft:** Lemi Belay Tolu, Wondimu Gudu.

**Writing – review & editing:** Lemi Belay Tolu, Wondimu Gudu.

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
