## [Decision Letter · Decision Letter 0]

19 Oct 2020

PONE-D-20-25230

Sexual assault cases at a tertiary referral hospital in urban Ethiopia: One-year retrospective review

PLOS ONE

Dear Dr. Tolu,

Thank you for submitting your manuscript to PLOS ONE. After careful consideration, we feel that it has merit but does not fully meet PLOS ONE’s publication criteria as it currently stands. Therefore, we invite you to submit a revised version of the manuscript that addresses the points raised during the review process.

You need to address all the issues raised by the referee, In particular the definitions of the terms used and the presentation of the data must be clarified.

We look forward to receiving your revised manuscript.

Kind regards,

Andrew R. Dalby, PhD

Academic Editor

PLOS ONE

Journal Requirements:

2. In the ethics statement in the manuscript and in the online submission form, please provide additional information about the patient records used in your retrospective study, including: a) whether all data were fully anonymized before you accessed them; b) the date range (month and year) during which patients' medical records were accessed; c) the date range (month and year) during which patients whose medical records were selected for this study sought treatment. If patients provided informed written consent to have data from their medical records used in research, please include this information. If the ethics committee waived the need for informed consent, please include this information.

Reviewers' comments:

Reviewer's Responses to Questions

**Comments to the Author**

1. Is the manuscript technically sound, and do the data support the conclusions?

Reviewer #1: Partly

2. Has the statistical analysis been performed appropriately and rigorously? 

Reviewer #1: N/A

3. Have the authors made all data underlying the findings in their manuscript fully available?

Reviewer #1: Yes

4. Is the manuscript presented in an intelligible fashion and written in standard English?

Reviewer #1: No

5. Review Comments to the Author

Reviewer #1: Sexual assault cases at a tertiary referral hospital in urban Ethiopia – PLOS ONE

Overall,

This study presents descriptive data on sexual assault cases within a single hospital in Ethiopia. The data are interested however, the paper is lacking in several major ways. 1) Sometime the language used is not clear; 2) it is unclear how this paper building on existing literature from this area; and 3) the question “so what” or “why does this matter” is not answered for this study. The findings need to be contextualized in terms of other literature and implications from the findings.

Abstract

• “Sexual assault is an important health and social problem affecting the young and less educated girls.” Please provide a citation from the “less educated” part of that sentence or remove.

• Forty (23.6%) = these numbers are not the same

• Spell out hrs.

• How can the average hours for reporting be greater than the top of the range (“with an average of 98 hrs.”)

• I don’t understand this sentence “cases respectively while urine pregnancy tests were done in 62.5% of the cases.” Nor this “Although it is largely hidden by the victims”

• Did you examine “forensic evidence collection”? If no, how can you make conclusion about this? If yes, please include in the results in the abstract.

Introduction

• “forced or coerced vaginal or anal penetration by any other body parts or object” is usually considered rape, not sexual assault. Double check your reference.

• What does “although it is largely hidden by the victims” mean? Do you mean, most victims do not disclose? If so, provide a reference.

• I do not understand this sentence “Reports from Ethiopia showed from a study of 367 high school girls, that 11.4% of them had started having intercourse and 33.3% of this group was rape” – just report on percent who were raped, not who were sexually active.

• For this sentence, the first part does not make sense with the second part of the sentence “Adolescents continue to have the highest rates of all age groups, assailants are known to their victims who perpetrate this act during the daytime and survivors often delay in seeking care”. Delete this part “Adolescents continue to have the highest rates of all age groups”

o Actually, this entire paragraph does not flow. It appears to be random sentence strung together without connections between them. Please link the ideas together.

• Your literature review only includes one article on sexual violence within Ethiopia. You need to justify a descriptive only paper by a literature review that indicates what other information has been published about Ethiopia and what information is not know (that you will present in your findings).

Methods and Materials

• Please indicate how many cases were excluded due to missing variables.

• I do not understand this sentence “The medical records of sexual assault at Michu clinic were approached for the identification of cases who received medical care within the study period as having experienced any form of sexual assault.”

• Instead of operational definitions, you should have a section on “measures” that tells the reader how each variable was defined.

o For example, what is the definition of a student? Primary? Secondary? either?

Tables

• Please organize the finding in a logical order – e.g., from most common to least common/largest to smallest percentage.

Results

• The results seem a bit unorganized. Please have a paragraph with highlighted findings from each table before each table. Some tables were barely referenced in the text (e.g., table 3) and other text appeared without a corresponding table (e.g., information starting with Among the 113 (66%) female victims who had body orifice penetration, genital penetration).

• For some findings, both the percentage and the sample n are given in the text. For others, just the %. Please include percentage and n throughout the writeup.

Discussion

• The discussion seems like a summary of the findings with little context to understand how these findings add to the existing literature. You need to highlight how these findings add to existing literature and are similar and different.

• There should be a section on implications (research, policy or clinical) for your findings. The question remains, why do these findings matter?

6. PLOS authors have the option to publish the peer review history of their article (what does this mean?). If published, this will include your full peer review and any attached files.

Reviewer #1: No

---

## [Author Response · Author response to Decision Letter 0]

30 Oct 2020

October 30/2020

Dear Editor in chief.

We would like to thank the reviewers and editor for their thoughtful review of the manuscript. They raised important issues and the inputs are very helpful for improving the manuscript. We agree with all comments and we have revised our manuscript accordingly. We respond below in detail to each of the raised comments. We hope that you find our responses satisfactory and that the manuscript is now acceptable for publication. 

I look forward hearing from you soon

Sincerely,

Lemi Belay Tolu (MD, Assistant prof of obstetrics and gynecology).

Saint Paul’s Millennium Medical College (SPHMMC)

Department of Obstetrics and Gynecology

Addis Ababa, Ethiopia.

Email: lemi.belay@gmail.com

Editor 

Response: Dear editor thank you very much, we adhered to PLOS ONE style requirements through manuscript preparation. 

2. In the ethics statement in the manuscript and in the online submission form, please provide additional information about the patient records used in your retrospective study, including: a) whether all data were fully anonymized before you accessed them; b) the date range (month and year) during which patients' medical records were accessed; c) the date range (month and year) during which patients whose medical records were selected for this study sought treatment. If patients provided informed written consent to have data from their medical records used in research, please include this information. If the ethics committee waived the need for informed consent, please include this information.

Response: Dear editor all data were anonymized, and written consent was waived. We included patient who sought care for sexual assault between October 1, 2018, up to October 1, 2019. We collected data for two months from December 1,2019 to January 30,2020. We modified the manuscript according to the comment as seen under ethical considerations.

Response: Dear editor thank you very much, Captains for Supporting Information files were incorporated at the end of the manuscript after the comment (Supporting information section, page 22, line 346)

Reviewer #1: Sexual assault cases at a tertiary referral hospital in urban Ethiopia – PLOS ONE

Overall,

This study presents descriptive data on sexual assault cases within a single hospital in Ethiopia. The data are interested however, the paper is lacking in several major ways. 1) Sometime the language used is not clear; 2) it is unclear how this paper building on existing literature from this area; and 3) the question “so what” or “why does this matter” is not answered for this study. The findings need to be contextualized in terms of other literature and implications from the findings.

Abstract

• “Sexual assault is an important health and social problem affecting the young and less educated girls.” Please provide a citation from the “less educated” part of that sentence or remove.

Response- dear reviewer less educated is removed than defining it in abstract section. 

• Forty (23.6%) = these numbers are not the same

Response: Dear reviewer original we mean number and its percentage equivalent and for better understanding we corrected by removing the number 

• Spell out hrs.

Response: done 

• How can the average hours for reporting be greater than the top of the range (“with an average of 98 hrs.”)

Response: dear reviewer the range is 2 hours to 93 days (2224 hours), average being 98 hours 

• I don’t understand this sentence “cases respectively while urine pregnancy tests were done in 62.5% of the cases.” Nor this “Although it is largely hidden by the victims”

Response: Dear reviewer the lengthy sentences was broken in to two” Serology tests for HIV, Hepatitis B, and Syphilis were done in 97.3%, 88.7%, and 84.5% cases respectively. Urine pregnancy tests were done in 62.5% of the cases”. The largely hidden is used to mean not reported and we corrected by replacing hidden by not reported for better understanding 

• Did you examine “forensic evidence collection”? If no, how can you make conclusion about this? If yes, please include in the results in the abstract.

Response: Dear reviewer vaginal swab collections are parts of forensic evidence for GBV but not adequate. It was with this consideration that we included the conclusion of inadequate forensic evidence collection. 

Introduction

• “forced or coerced vaginal or anal penetration by any other body parts or object” is usually considered rape, not sexual assault. Double check your reference.

Response: Dear reviewer comment well taken, and the sentences corrected as sexual assault and rape spectrum to include all.

• What does “although it is largely hidden by the victims” mean? Do you mean, most victims do not disclose? If so, provide a reference.

Response: Dear reviewer thanks for the concern we have provided the reference. 

• I do not understand this sentence “Reports from Ethiopia showed from a study of 367 high school girls, that 11.4% of them had started having intercourse and 33.3% of this group was rape” – just report on percent who were raped, not who were sexually active.

• For this sentence, the first part does not make sense with the second part of the sentence “Adolescents continue to have the highest rates of all age groups, assailants are known to their victims who perpetrate this act during the daytime and survivors often delay in seeking care”. Delete this part “Adolescents continue to have the highest rates of all age groups”

o Actually, this entire paragraph does not flow. It appears to be random sentence strung together without connections between them. Please link the ideas together.

Response: Dear reviewer we really appreciated this very important input. We rearranged as. Reports from Ethiopia showed that from 367 high school girls, about 33.3% of the participants first intercourse was rape. The following sentences was deleted: “Adolescents continue to have the highest rates of all age groups”

• Your literature review only includes one article on sexual violence within Ethiopia. You need to justify a descriptive only paper by a literature review that indicates what other information has been published about Ethiopia and what information is not known (that you will present in your findings).

Response: Dear reviewer we included more evidences from Ethiopia in the introduction section. (Introduction, page 3, lines 54-59)

Methods and Materials

• Please indicate how many cases were excluded due to missing variables.

Response: Dear reviewer we have provided this information in the result section of the manuscript. (Result, Page 7, line 157)

• I do not understand this sentence “The medical records of sexual assault at Michu clinic were approached for the identification of cases who received medical care within the study period as having experienced any form of sexual assault.”

Response: Dear reviewer words were added for clarification and corrected as “The medical records of sexual assault cases managed at Michu clinic were approached for the identification of alleged sexual assault cases who received medical care within the study period” (Page 5, line 91)

Results

Tables

• Please organize the finding in a logical order – e.g., from most common to least common/largest to smallest percentage.

• The results seem a bit unorganized. Please have a paragraph with highlighted findings from each table before each table. Some tables were barely referenced in the text (e.g., table 3) and other text appeared without a corresponding table (e.g., information starting with Among the 113 (66%) female victims who had body orifice penetration, genital penetration).

• For some findings, both the percentage and the sample n are given in the text. For others, just the %. Please include percentage and n throughout the writeup.

Response: Dear reviewer, though it is difficulty to arrange from largest to smallest based on results as we presented based on logical order of variables, we have reorganized the tables. We rearranged also table and description pattern including their order. Number and percentage were used in the description as per the comment (Result section)

Discussion

• The discussion seems like a summary of the findings with little context to understand how these findings add to the existing literature. You need to highlight how these findings add to existing literature and are similar and different.

• There should be a section on implications (research, policy or clinical) for your findings. The question remains, why do these findings matter?

Response: Dear reviewer we revised the discussion section as per the comment by omitting more of things that look summary of findings and included literature contexts. Conclusion section was reorganized as subsection with clinical implications and research and policy implications (Discussion and Conclusion section).

---

## [Editor Report · Decision Letter 1]

20 Nov 2020

Sexual assault cases at a tertiary referral hospital in urban Ethiopia: One-year retrospective review

PONE-D-20-25230R1

Dear Dr. Tolu,

We’re pleased to inform you that your manuscript has been judged scientifically suitable for publication and will be formally accepted for publication once it meets all outstanding technical requirements.

Kind regards,

Andrew R. Dalby, PhD

Academic Editor

PLOS ONE
---

## [Editor Report · Acceptance letter]

26 Nov 2020

PONE-D-20-25230R1 

Sexual assault cases at a tertiary referral hospital in urban Ethiopia: One-year retrospective review 

Dear Dr. Tolu:

I'm pleased to inform you that your manuscript has been deemed suitable for publication in PLOS ONE. Congratulations! Your manuscript is now with our production department. 

Kind regards, 

on behalf of

Dr. Andrew R. Dalby 

Academic Editor

PLOS ONE